# Pursuing the Elixir of Life: In Vivo Antioxidative Effects of Manganosalen Complexes

**DOI:** 10.3390/antiox9080727

**Published:** 2020-08-10

**Authors:** Lara Rouco, Ana M. González-Noya, Rosa Pedrido, Marcelino Maneiro

**Affiliations:** 1Departamento de Química Inorgánica, Facultade de Ciencias, Universidade de Santiago de Compostela, 27002 Lugo, Spain; lara.rouco.mendez@usc.es; 2Departamento de Química Inorgánica, Facultade de Química, Universidade de Santiago de Compostela, 15782 Santiago de Compostela, Spain; ana.gonzalez.noya@usc.es (A.M.G.-N.); rosa.pedrido@usc.es (R.P.)

**Keywords:** reactive oxygen species (ROS), oxidative stress, catalytic antioxidants, superoxide dismutase, catalase, peroxidase, manganese, salen-type ligands, animal studies

## Abstract

Manganosalen complexes are coordination compounds that possess a chelating salen-type ligand, a class of bis-Schiff bases obtained by condensation of salicylaldehyde and a diamine. They may act as catalytic antioxidants mimicking both the structure and the reactivity of the native antioxidant enzymes active site. Thus, manganosalen complexes have been shown to exhibit superoxide dismutase, catalase, and glutathione peroxidase activities, and they could potentially facilitate the scavenging of excess reactive oxygen species (ROS), thereby restoring the redox balance in damaged cells and organs. Initial catalytic studies compared the potency of these compounds as antioxidants in terms of rate constants of the chemical reactivity against ROS, giving catalytic values approaching and even exceeding that of the native antioxidative enzymes. Although most of these catalytic studies lack of biological relevance, subsequent in vitro studies have confirmed the efficiency of many manganosalen complexes in oxidative stress models. These synthetic catalytic scavengers, cheaper than natural antioxidants, have accordingly attracted intensive attention for the therapy of ROS-mediated injuries. The aim of this review is to focus on in vivo studies performed on manganosalen complexes and their activity on the treatment of several pathological disorders associated with oxidative damage. These disorders, ranging from the prevention of fetal malformations to the extension of lifespan, include neurodegenerative, inflammatory, and cardiovascular diseases; tissue injury; and other damages related to the liver, kidney, or lungs.

## 1. Introduction

In 2000, a research paper by Melov et al. [1] reported the extension of the lifespan of nematode worms (*Caenorhabditis elegans*) by treatment with different manganosalen complexes which act as synthetic scavenger compounds. This study planned to test the theory that reactive oxidative species cause aging. In announcing this research, Melov stated that “the results are the first real indication we have that aging is a condition that can be treated through appropriate drug therapy” [2]. The research paper and this statement attracted extensive coverage in communication media, where these findings were presented as a sort of search for the elixir of life. The published results indicate that the two tested compounds (named as EUK-8 and EUK-134) increased the mean lifespan of the worms by 44 percent over the control group and that treatment of prematurely aging worms resulted in normalization of their lifespan, which means a 67 percent increase. However, the chosen nematode worm is a species that has 959 cells in its body whereas humans have 100 trillion, constituting one of the weaknesses of animal models for translational research [3]. To investigate the protective activity of the manganosalen catalytic antioxidants in a mammalian animal model, Melov et al. used mice lacking SOD2, the mitochondrial form of superoxide dismutase [4]. The genetically engineered SOD2 nullizygous mice lack the oxygen-scavenging enzyme that helps protect mitochondria from free radicals, and they usually die within the first week of life by suffering different pathologies. When the mice were injected with the manganosalen complexes (EUK-8, EUK-134, and the compound named as EUK-189), they lived three times longer, indicating that these artificial complexes may act as synthetic mimetics of SOD in an animal model lacking critical antioxidant functions and, therefore, without direct relevance to the natural lifespan. In this study, the manganosalen complexes also rescued mice from oxidative neurodegenerative process, and the results suggest that this class of synthetic catalytic antioxidants can permeate the brain, can gain access to the mitochondria, and can attenuate mitochondrial damage attributable to oxidative stress.

Experimental approaches have yielded contradictory evidences about the therapeutic attenuation of aging using these synthetic compounds [5], which will be discussed further below. However, a significant number of in vivo experimental studies have shown that these complexes exhibit remarkable efficacy in several animal models suffering from oxidative stress injuries [6,7,8,9,10]. The most common animal model is that of rats and different types of mice, but in addition to these and the mentioned nematodes, studies were also carried out on pigs, sheep, or fish or by using the vinegar fly. Additionally, there are also studies of the protective effect against induced oxidative stress in other models than animals, like *E. coli* [11], and in humans, with an evaluation of the beneficial effects over ultraviolet A-exposed skin in vivo [12].

After more than two decades of in vivo studies using manganosalen complexes, the focus of this review is on the evaluation of their effectiveness on the treatment of several pathological disorders associated with oxidative damage: aging; neurodegenerative diseases and mental disorders; inflammatory, cardiovascular, or liver diseases; skin damage; fetal malformations; and other damages related to the kidney or lungs. Moreover, this review collects recent advances in the state of knowledge on the molecular mechanisms for the antioxidant activity of these compounds [13,14,15,16,17,18,19,20,21,22,23].

## 2. Manganese Superoxide Dismutases, Peroxidases, and Catalases

Reactive oxygen species (ROS) are partially reduced metabolites of molecular oxygen formed in biological systems as a result of a normal cellular metabolism, primarily in the mitochondria [24]. Exogenous sources such as pollutants [25], smoke [26], radiation [27], or heavy metals [28] also may increase the ROS levels. Superoxide radical anion (O_2_^•−^), hydrogen peroxide (H_2_O_2_), hydroxyl radical (OH^•^), lipid hydroperoxide (LOOH), lipid radical (L^•^), peroxyl radical (ROO^•^), peroxynitrate (ONOO^−^), and other radicals which can be produced through a sequence of reactions constitute the designated ROS. Free radicals are generated when oxygen interacts with certain molecules, leading to highly reactive species with one or more unpaired electron(s). ROS may play a role in signaling functions, consequently activating protective and adaptative programs [29,30]. Nevertheless, excessive ROS levels cause oxidation of organic molecules with alteration of their structure and biological functions, activation of phagocytes, release of cytokines, or activation of oncogenes [31,32,33,34,35]. These processes lead to different pathologies in humans, such as carcinogenesis [36,37], inflammatory illnesses [38], diabetes type II [39], cellular senescence [40], and different neurodegenerative diseases [41,42].

ROS levels are regulated in the living systems by the antioxidant enzymes, which basically include superoxide dismutase (SOD), catalase (CAT), glutathione peroxidase (GPx), peroxiredoxins (PRDXs), thioredoxin 2 (TRX2), or cytochrome c oxidase (complex IV). Other nonenzymatic antioxidants such as α-tocopherol, ascorbic acid, and carotenes complete the antioxidant defense grid [43]. In this review, the discussion will be restricted to the antioxidant enzymes in which the function can be mimicked by manganosalen complexes: SOD, CAT, and GPx (Figure 1). The global role of these cellular antioxidant defenses is the reduction of ROS to water.

Superoxide radical anion is a potential harmful species produced during respiration from the one-electron reduction of molecular oxygen. SOD catalyzes the dismutation of two superoxide anions to yield hydrogen peroxide and molecular oxygen [44]. SODs are classified according to their metal ion cofactor as Fe-SODs, Ni-SODs, CuZn-SODs, and MnSODs [45]. SOD2, also known as manganese-dependent superoxide dismutase, is located within the inner mitochondrial matrix, the main site of free radical production from the electron transport chain [46], being pivotal in ROS release in humans. The coordination environment around the manganese ion for human SOD2 is depicted in Figure 1. The superoxide disproportionation mechanism by SOD2 involves a cyclic one-electron oxidation and reduction of the manganese ion between Mn(II)/Mn(III) oxidation states (Figure 2). The catalysis occurs at a rate approaching the diffusion-controlling region (log K_SOD_ ~ 10^9^ M^−1^ s^−1^) [47].

Hydrogen peroxide is generated as a by-product of mitochondrial electron transport of aerobic respiration. As it has been mentioned, it is one of the products of the superoxide radical anion dismutation by SOD. Although H_2_O_2_ is less reactive than superoxide, control of H_2_O_2_ levels is also critical [48]. Catalase enzymes catalyze the decomposition of hydrogen peroxide to water and oxygen [49]. The non-heme dinuclear Mn catalase was isolated from bacteria. Figure 1 shows the core of the active site of this enzyme in *Lactobacillus plantarum* [50]. The catalase reaction proceeds via a two-electron oxidation-reduction cycle during turnover (Figure 3).

Peroxidases catalyze the oxidation of a broad range of substrates by hydrogen peroxide [51]. Hydrogen peroxide is then reduced by two electrons using glutathione (GSH) as a sacrificial reductant, yielding water and glutathione disulfide (GSSG) (Figure 4). The manganese binding site of the manganese peroxidase is also shown in Figure 1 [52]. Peroxidases are not limited to H_2_O_2_ as the substrate but also catalyze the conversion of organic peroxide to alcohol. Thus, peroxidases can eliminate lipid hydroperoxides, another ROS which contributes to the progression of disbalanced redox homeostasis [53]. Lipid hydroperoxides can be generated very easily in neuronal membranes rich in polyunsaturated fatty acids like arachidonic acid, docosahexaenoic acid, or eicosapentaenoic acid.

The enzymatic antioxidant defense along with other nonenzymatic antioxidants act in an effective way against free radicals and other reactive oxygen species to control their damaging effects to macromolecules and body tissues. However, overproduction of ROS leads to oxidative damage of cellular structures due to an imbalance in the oxidant-antioxidant status [54]. ROS excess may be derived from exogenous sources as mentioned above and by faulty regulation of cellular antioxidant defenses, normally associated with aging, that can lead to accumulation of toxic levels of ROS [55]. For this situation, a strategy to recover a balance between ROS generation and removal is the use of antioxidants as therapy [56,57,58,59]. Administration of exogenous native antioxidant enzymes has not been successful for therapeutic treatment of oxidative stress because of several limitations [60]: (i) short half-life of the enzymes; (ii) difficulty to enter in the cells due to their high molecular weight; (iii) antigenicity; and (iv) high-manufacturing costs. To overcome these limitations, pharmacological research has pointed at the development of low molecular weight SOD/CAT mimics [61,62,63,64]. Several supplements as α-tocopherol [65], ascorbic acid [66], carotenes [67,68], or other organic molecules [69,70,71,72] with antioxidant properties have been also used for the therapeutic treatment of oxidative stress-induced diseases.

## 3. Manganosalen Complexes as Catalytic Antioxidants

Manganosalen complexes are coordination compounds with a chelate bis-Schiff base ligand obtained by condensation of salicylaldehyde and a diamine. The acronym *salen* is due to the reactants used to synthesize the ligand N,N-bis(salicylidene)ethylendiamine, obtained by condensation of salicylaldehyde (sal) and ethylenediamine (en) [73]. Although the latter ligand is strictly speaking the salen ligand, actually the class of compounds known as manganosalen complexes encompasses other bis-Schiff base ligands (containing two bisimine groups), obtained from different diamines (propylenediamine, phenylenediamine, butylendiamine, etc.) as well as the group of their derivatives with different substituents both in the phenyl rings and in the diamine spacer [20,74]. Figure 5 collects some manganosalen complexes with pharmacological relevance used in in vivo studies, corresponding to the EUK series patented by Eukarion.

This type of ligand has oxygen and nitrogen donor groups, which decrease the Mn(III)/Mn(II) redox potentials upon coordination (E° = 1.51 V for [Mn(H_2_O)_6_]^3+^, a rather oxidizing potential), and the resulting complexes constitute suitable systems to catalyze multiple redox reactions [20,74,75,76,77,78]. Redox potentials of manganosalen complexes may be tuned by modifying features other than these substituents. In this sense, the number of ligands or the geometry around the metal ion are factors that influence the manganese redox potential of the final complex. For example, alkoxy substituents in the phenyl rings, particularly 3-methoxy or 3-ethoxy groups (see EUK-113, EUK-134, EUK-172, EUK-178, or EUK-189 in Figure 5), lower the redox potentials for the manganosalen complexes due to the electron-donor character of these substituents, thus facilitating achievement of higher oxidation states for the manganese ion during enzymatic activity.

The chelating bis-Schiff base ligand forms a typical almost planar MnN_2_O_2_ core where the 5- or 6-member chelate rings confer high stability. The introduction of different auxiliary ligands (halides, carboxylates, alcohols, dicyanamides, thiocyanates, etc.) alters the geometry of the compounds [20,79], giving rise to different behaviors in catalysis. Labile auxiliary ligands or solvent molecules in the axial positions favor catalytic activity through an inner-sphere electron transfer mechanism in which a vacant can be generated in this site, where the substrate molecule can be subsequently accommodated [80]. A short two-carbon chain between the imine groups in the Schiff base ligand constricts the chelate ring once the nitrogen atom coordinates to the metal, leading to tetragonally elongated geometries [23,80,81,82,83]. On the contrary, if the Schiff base ligand has a flexible three-membered alkyl chain between the imine groups, a better stabilization of a high-symmetry octahedral symmetry is achieved and, subsequently, the generation of a coordination site gets difficult. Thus, a correlation between the factor of the tetragonal elongation and catalytic activity as enzyme mimics has been reported for manganosalen complexes [23,80,81,82,83].

Supramolecular interactions may play different crucial roles in enhancing the activity as enzyme mimics. On the one hand, hydrogen bonding and other supramolecular contacts may induce self-organization of the complexes to afford dimeric entities [84]. Thus, these supramolecular mechanisms may allow aggregation of the complexes into dimers once the monomers cross the cell membrane. On the other hand, supramolecular interactions also play an essential role in biological processes recognition [85]. Second-sphere effects of the substituents may also modulate the redox potentials of the metal center and the metal–ligand bond strength and may guide their reactivity with superoxide anion radical [86]. Alkoxy substituents on the phenyl rings can also participate in establishing supramolecular interactions through hydrogen bonding, which added to their mentioned effect on redox potentials could explain the high activity as enzyme mimics of alkoxy substituted manganosalen complexes [87,88,89].

The relatively planar MnN_2_O_2_ core of manganosalen complexes is somewhat similar to that of natural manganese Mn-macrocycles or Mn-porphirins, and this similarity is shown in their chemical reactivity. However, the manganese ion in manganosalen compounds is coordinated to oxygen and nitrogen atoms, which contrasts with porphyrins where the metal is coordinated to nitrogen atoms only. Oxygen and nitrogen are the most common donor atoms in biological systems, so the structure of manganosalen complexes (Figure 6) resembles those of various native manganoenzymes. Manganosalen compounds are easily obtained from inexpensive precursors, and they are cheaper than Mn-porphirins.

One of the most interesting properties of the manganosalen complexes is that they are cell-permeable, with better bioavailability than exogenous antioxidant enzymes [90,91]. Manganosalen complexes exhibit high SOD, catalase, and peroxidase activities, which has led to their development as catalytic antioxidants [6,7,8,9,10,13,18,19,20,21,23], a term used for all cases in which one single molecule of catalyst induces the detoxification of numerous ROS molecules.

Malfroy et al. first reported the SOD mimetic properties of manganosalen complexes [92]. Based on stopped-flow analysis combined with time-resolved UV/vis spectroscopy and global spectra analysis of superoxide decay, it has been reported that manganosalen complexes possess a SOD activity of about 2 × 10^6^ M^−1^ s^−1^ [93,94], both in Hepes and in phosphate buffers. The mechanism followed by the manganosalen complexes is similar to that of the native SOD enzyme, a ping-pong mechanism where a superoxide anion radical reduces the synthetic manganese complex from Mn(III) to Mn(II), which is subsequently oxidizes back to Mn(III) by a second superoxide anion radical (Figure 7a) [10,19,23].

While the SOD activity of manganosalen complexes hardly varies with derivatization, their catalase activity is highly sensitive to the substituents on the aromatic rings or the length of the alkyl chain in the spacer between the imine groups [89], ranging from inactivity to high efficiencies in the hydrogen peroxide disproportionation [80,81,82,83,95]. The efficiency of these systems seems to be related to the presence of at least one vacancy or a labile coordination position on the manganese core. Electron donor substituents on the phenyl rings of the salen moiety also increase the catalase activity of manganosalen complexes [86]. The rates at which manganosalen complexes scavenge hydrogen peroxide are similar to those reported for metalloporphyrins. For instance, compound EUK-172 (see Figure 5) is reported to have a catalase rate greater than 1 mM O_2_/min [95], measured by monitoring the conversion of hydrogen peroxide to oxygen using a Clark type oxygen electrode [96].

The mechanism of these synthetic mimics is proposed to involve mononuclear Mn(V)=O species [97,98] (Figure 7b), although some mimics may follow a mechanism through the formation of dimeric species in solution [80]. Anyhow, the mechanism is different from the native catalase enzyme shown in Figure 3.

Peroxidase mimics are able to convert H_2_O_2_ to H_2_O and to scavenge other peroxides, including lipid hydroperoxides or other organic peroxides [95,99], so that they could scavenge lipid peroxides in tissues. Manganosalen complexes are proposed to behave as peroxidase mimics by a mechanism quite similar to that of their catalase action, through a Mn(V)=O intermediate, which is able to oxidize an organic substrate (Figure 7b), to afford the initial catalyst complex.

Manganosalen complexes often have multiple antioxidant activities at the same time, showing, for instance, both SOD and catalase functions. This reactivity against different ROSs is attractive since hydrogen peroxide is a product released by SOD activity.

## 4. Therapeutic Effects of Manganosalen Complexes in In Vivo Models

The SOD, catalase, and peroxidase activities shown by manganosalen complexes have attracted attention for their use as catalytic antioxidants. Subsequent in vitro studies have confirmed their efficiency in oxidative stress models [6,8,9,10,13,14,15,20,21,23] and other references cited therein, although the focus of this review is on the evaluation of the in vivo studies with different animal models. These studies are organized below according to the pathology or the oxidative damage produced by ROS.

### 4.1. Neurodegenerative Diseases and Mental Disorders

The brain is prone to oxidative stress as a result of the high levels of oxygen required, which represents 20% of oxygen uptake when brain accounts for only 2% of body weight. Neurons consume a high rate of energy (4 × 10^12^ ATP/minute), meaning large amounts of oxygen to maintain neural intracellular ion homeostasis [41]. Moreover, neural membranes have high concentrations of polyunsaturated fatty acids which generate lipid hydroperoxides. Additional factors, as the presence of auto-oxidizable neurotransmitters, increase the sensitivity of this organ to ROS-mediated damage. Excessive ROS levels have been associated with neurodegenerative disorders like Alzheimer’s Disease (AD), Parkinson’s Disease (PD), amyotrophic lateral sclerosis (ALS), or Huntington’s Disease [42].

As previously mentioned, the EUK-8, EUK-134, and EUK-189 models were injected to SOD2 nullizygous mice to rescue them from oxidative neurodegenerative process [4]. These compounds had previously shown efficacy in a variety of oxidative stress paradigms [100,101,102]. For instance, EUK-134 had been found to protect most of the vulnerable neurons from excitotoxic cell death in Sprague–Dawley rats [101]. Administration of these three manganosalen compounds extended the lifespan of the mice lacking SOD2 by approximately threefold and eliminated clinical signs of the associated neurobehavioral phenotype previously described. This study also indicated that these catalytic antioxidants could cross the blood–brain barrier, particularly EUK-189, which is slightly more lipophilic than EUK-134. In a later study using the same animal model, Melov et al. reported that neural cell death in defined regions of the frontal cortex of SOD2 null mice is a consequence of endogenous mitochondrial oxidative stress [103]. In this study, they could partially rescue neural cell death by treatment with a high dose of EUK-8. Melov et al. also reported the therapeutic effects of EUK-189 in preventing a neurodegenerative phenotype in an Ah-transgenic mouse model for Alzheimer’s disease [104]. This manganese complex and the cyclic analogue EUK-207 were used in a study with C57BL/6N Sim middle-age mice [105], which usually exhibit a dramatic decrease in learning and memory function between 8 and 11 months of age, associated with oxidative protein damage in the brain [106]. Treatment during a 3-month period with EUK-189 or EUK-207 resulted in an almost complete reversal of age-related learning and memory deficit, showing a complete reversal in protein oxidation and a 50% reduction in age-related increase in lipid peroxidation. EUK-207 exhibits longer plasma half-life than EUK-189 and subsequently greater biological stability. Two such molecules were also used by Baudry et al. in older mice, at a lower dose, and for longer periods of time [107], preventing age-dependent cognitive decline in the same way as previously found for middle-aged individuals. Their results indicate that the age-associated deficits in learning and memory might be originated by oxidative damage to hippocampus, amygdala, or both.

EUK-8 was the first manganosalen complex that showed efficacy for the treatment of a neurodegenerative disease using an animal model (see Table 1). Malfroy et al. reported in 1997 [102] how this synthetic catalytic scavenger reduces the severity of autoimmune encephalomyelitis in guinea pigs. Watanabe et al. [91] used the same compound to treat small bowel ischemia/reperfusion injury in Sprague–Dawley rats. They compared the protective effects of EUK-8 and a Mn(III)-porphirin SOD mimic, manganese-meso-tetrakis(N-methylpyridinium-2-yl)porphyrin [108]. Both the manganosalen compound and Mn-porphyrin showed similar beneficial properties by the inhibition of O_2_, H_2_O_2_, and NO production.

Xu et al. [109] reported that EUK-8 and EUK-134 reduced the levels of oxidative stress and prolonged survival in a mouse amyotrophic lateral sclerosis model. In a later study, EUK-134 afforded the best results compared to EUK-8 in reducing brain infarct size after middle artery occlusion in a rat model [110]. After the neuroprotective effects shown by EUK-8 and EUK-134, Doctrow et al. [85] compared the in vivo antioxidant activity of different analogues, including the two cited compounds, EUK-113, EUK-161, EUK-163, EUK-172, EUK-178, and EUK-189 in a rat stroke model. They concluded that alkoxy substituents on the phenyl rings for this series of compounds confers some advantage toward the biological protective effects of these manganosalen complexes.

Andersen et al. [111] demonstrated the efficacy of EUK-134 and EUK-189 in protecting against paraquat-induced dopaminergic cell death in adult mice via inhibition of the activation of Jun N-terminal kinases (JNK)-mediated apoptosis. Paraquat is an herbicide that induces selective loss of dopaminergic neurons of the substantia nigra. These two manganosalen complexes significantly inhibited caspase-3 activation, cell death, and DNA fragmentation in in vitro paraquat exposed rat cells. EUK-189 was also employed in a mouse model of human prion disease [112], giving place to a modest 5% increase of survival compared to untreated disease controls. This beneficial effect was correlated with reductions in oxidative and, especially, nitrative damage to proteins. The same manganosalen complex, EUK-189, was used by Levine et al. [113] to treat the neurobehavioral defect in ataxia-telangiectasia mice, showing that this catalytic antioxidant corrected the neurobehavioral abnormality. This effect, that was reproducible over time, involved a reduction in the oxidation of brain fatty acids and a retarded development of thymomas.

More recent studies were focused on the manganosalen EUK-207 due to both its ability to suppress oxidative stress and its greater biological stability. Thus, Raber et al. [114] reported that this compound mitigated radiation-induced cognitive impairments without affecting cognition of sham-irradiated mice. Wang et al. [22,115] used EUK-207 to treat hydrogen peroxide-induced DNA damage and senescence phenotype in senescence-accelerated mouse-prone 8 mice, reducing age-related loss of both hearing and hair cell degeneration. In their study, they found that cochleae cells treated with EUK-207 displayed increased levels of FOXO3a and Nrf2, two transcription factors that have been previously shown to positively regulate cellular resistance to oxidative stress [116,117].

### 4.2. Inflammatory Diseases

The neuroprotective effects shown by manganosalen complexes are close related to their ability to attenuate inflammation makers such as cytokines or chemokimes. High amounts of hydrogen peroxide activate a number of transcription factors as NFkB, AP-1, and Nrf2 [118], for which the levels may be regulated by the administration of manganosalen complexes. The anti-inflammatory effects of these catalytic antioxidants play a beneficial role not only in neurons or in the respiratory system but also in a wide range of pathologies related to inflammation (Table 2).

Hill et al. [119] treated Sprague–Dawley rats with EUK-189 to mitigate DNA damage in rat lungs after exposure to 10–20.5 Gy doses of gamma rays. The catalytic antioxidant was effective at reducing micronucleus formation in lung fibroblasts, and this could be attributed to the ability of this compound to suppress the signal that would normally turn on the inflammatory response to repair radiation insult.

EUK-134 was used by Lawler et al. [120] as an ROS scavenger in myopathy in the diaphragm of the *mdx* mouse model. This manganosalen complex reduced hydroperoxides, markers of oxidative stress in this muscle, attenuated both by the elevation of inflammatory cell invasion and by the NF-κB activity and p65 subunit protein levels in the *mdx* diaphragm. Muzykantov et al. [121] employed polyethylene glycol (PEG)-liposomes loaded with EUK-134 to alleviate acute pulmonary inflammation induced by endotoxin in mice. This manganosalen compound was also used by Singh et al. [122] to mitigate zinc- and paraquat-induced toxicity in rat polymorphonuclear leukocytes.

Other manganosalen compounds were used with efficacy against inflammatory episodes is EUK-207. Hill et al. [123] reported that this synthetic model mitigated the radiation-induced lung injury in 6- to 7-week-old Fisher rats by scavenging ROS and by reducing activity of the NFkB pathway. In this study, EUK-207 also showed the ability to reduce levels of TGF-β1 expression, activated macrophages, and fibrosis.

A study that deserves a more detailed analysis is the one carried out by Kash et al. [124] with EUK-207 to reduce lung damage and to increase survival during 1918 influenza virus infection in mice. This pandemic, caused by the H1N1 influenza A virus, infected about a third of the world’s population of 1918 and resulted in 40–60 million deaths worldwide. This influenza and SARS-CoV-2 (severe acute respiratory syndrome coronavirus 2) share some similarities in the way that they may lead to respiratory failure. Many studies demonstrated that the severe lung pathology provoked by the 1918 influenza virus infection was associated with immunopathogenic immune response with excessive inflammatory and cell death responses [125]. EUK-207 treatment caused a marked reduction in the severity of lung pathology and substantially reduced cell death responses at both the RNA and the protein levels. The beneficial effect of this manganosalen complex in an animal model affected by the 1918 influenza is related to its ability to reduce the most relevant cytotoxic effects of ROS, thereby limiting excess cell death responses and allowing for increased lung repair responses. As the antioxidant effects of EUK-207 regulate the response of the host, it could be a therapeutic alternative to other organism’s infections compared to influenza. In this sense, it would be interesting to test this hypothesis with studies in animal models of the response to combined treatments of manganosalen complexes and antiviral against the SARS-CoV-2 virus.

Different previous studies showed the beneficial effects of manganosalen complexes to treat lung injuries. EUK-8 was used by Fink et al. to attenuate many of the features of lipopolysaccharide (LPS)-induced acute lung injury in a porcine model [96,126,127] by detoxifying ROS without affecting the release of other important proinflammatory mediators like 6-keto-prostaglandin F1 alpha, thromboxane B2, or tumor necrosis factor alpha. EUK-134 was the antioxidant catalytic model chosen by Kamp et al. [128] to alleviate asbestos- and H_2_O_2_-induced damage in mice, limiting pulmonary fibrosis. Hill et al. [129,130] reported the mitigation of radiation-induced lung injury by EUK-207 in Sprague–Dawley rats. Treatment with this manganese-model decreased hydroxyproline content, 8-hydroxy-2-deoxyguanosine, malondialdehyde levels, and activated macrophages levels. Lung levels of the cytokine transforming growth factor-β1 also decreased. Medhora et al. [131] reported the antioxidant effect of EUK-207 to reduce pneumonitis and pulmonary fibrosis after thoracic irradiation in a rat model.

The anti-inflammatory effect of other catalytic antioxidants, inspired by manganosalen complexes but not belonging to the EUK series, has also been studied. A bioinspired manganese SOD mimic, reported by Policar et al. [132], demonstrated efficiency as an anti-inflammatory agent for C57BL/6 mice with dinitrobenzene sulfonic acid (DNBS)-induced colitis.

### 4.3. Cardiovascular Diseases

Inflammation and tissue damage are closely related to different pathologies, including some cardiovascular diseases. Inflammation is common for heart disease and stroke patients. The role of excessive ROS production during hemorrhagic shock and reperfusion injury has been well documented [133]. In this way, some already cited studies of manganosalen complexes treatments for anti-inflammatory models also showed beneficial effects for cardiovascular diseases (Table 3). For instance, the commented in vivo study of Medhora et al. [131] with EUK-207 mitigated multiple vascular injuries in irradiated lungs.

De Windt et al. [134] used EUK-8 to reduce cardiac oxidative stress in Harlequin mutant mice and their wild-type counterparts. The results of this study showed an improvement of the left ventricular end-systolic dimensions and a fractional shortening by using this manganese complex. EUK-8 also attenuated necrotic and apoptotic cell death, prevented myocardial oxidant stress, and attenuated cardiac hypertrophy and fibrosis.

EUK-189 increased 30-day survival in irradiated mice according to a study by Whitnall et al. [135]. This manganosalen complex increased the number of diverse circulating blood elements like total white blood cells, lymphocytes, eosinophils, and platelets. The same catalytic antioxidant was tested by Monsalve et al. [136] to regulate ROS homeostasis and to control the vascular endothelial cells function in mice. EUK-189 restored endothelial growth factor-A signaling in peroxisome proliferator-activated receptor γ co-activator 1α (PGC-1α), a process to be relevant in metabolic disorders where microvascular complications are frequent, like diabetic retinopathy. Excessive ROS appeared as key factor in the alteration of the endothelial growth factor-A signaling and in the capacity of endothelial cells to form stable interactions with other endothelial cells and with the extracellular matrix, but these alterations were partially reversed by administration of EUK-189.

Recent studies have shown the efficacy of other manganosalen complexes for the treatment of different cardiovascular models. Yamada el al. [137] reported that EUK-134 prevented the force decrease and the actin modifications in pulmonary hypertension diaphragm bundles in Wistar rats. In their study, they found that this manganosalen complex does not alter diaphragm contractile function in normal rats. Lindner et al. [138] evaluated the therapeutic effects of EUK-207 in mice with age-dependent atherosclerosis. Long-duration therapy (40 weeks) with EUK-207 almost completely suppressed plaque development and macrophage content in thoracic aorta of the treated mice compared with control mice. However, therapy for eight weeks did not affect the area or the macrophage content.

### 4.4. Skin Damage

In the same way as discussed for endothelial cells, any tissue, including the skin, can suffer oxidative damage both of inflammatory and non-inflammatory origins. Several studies have focused on the protective effects of the manganosalen compounds in ROS-mediated skin damage (Table 4). Skin is exposed to solar ultraviolet irradiation, ozone, smoke, and air pollution. All of them are environmental sources of ROS that induce damage to lipids, proteins, and DNA, playing a role in the skin aging process [139].

EUK-8, EUK-134, and EUK-189 were used by Benichou et al. [140] to delay the rejection of fully allogeneic skin transplants in mice. Mice treated with EUK-189 showed the longest skin graft survival and, along with EUK-134, exhibited the longest delays of graft rejection. The three manganosalen complexes reduced anti-donor cytotoxic responses in skin-grafted mice, and they decreased pro-inflammatory type 1 alloresponse while promoted anti-inflammatory type 2 alloimmunity.

Declercq et al. [12] carried out a study of the protective effects of EUK-134 on the human skin of 748 healthy volunteers (18–80 years of age) over a period of 4 years. EUK-134 had been previously reported to increase cell survival in normal human keratinocytes upon exposure to ultraviolet-B, superoxide, or hydrogen peroxide [141,142]. In the study with the human volunteers, EUK-134 (applied at a concentration of 0.01–0.1%) reduced the level of skin surface lipid peroxidation in UVA-exposed skin. Noteworthily, the reduction of squalene hydroperoxide levels at the skin surface was found even when applying the antioxidant after UVA exposure. As a consequence of this study, EUK-134 is now commercially available as an antioxidant for the protection of dry or irritated skin.

EUK-207 was tested in a study performed by Lazar et al. [143] as a potential mitigating drug on end points relevant to radiation dermatitis, skin wound healing, and chronic oxidative stress in rats. The EUK-207-treated mice group showed reduced radiation dermatitis severity by 30 days after irradiation and displayed significantly smaller wounds than vehicle-treated rats. This manganosalen complex also reversed and normalized the gene expression pattern in irradiated skin by reducing the oxidation of proteins and nucleic acids. The same compound was used by Hill et al. [144] to mitigate the radiation-induced DNA damage and the lipid peroxidation in mice. They found that EUK-207 provided some protection against DNA damage only when delivered before irradiation. They also demonstrated significant protecting effects on radiation-induced lipid peroxidation at one or more of the three time points after local skin irradiation.

### 4.5. Fetal Malformations

Pregnancy is a state of oxidative stress due to high metabolic activity in the fetoplacental compartment. Regulation of ROS during gestation is a complex process, whereas excessive oxidant levels cause biomolecules damage and leads to fetal malformations as a consequence of the attack by ROS formed during the resumption of placental perfusion. On the other hand, the maintenance of a physiological level of oxidant levels is essential for governing life processes through redox signaling [145]. Two studies have been reported about the fetal protection or the reduction of pregnancy complications by manganosalen complexes.

Zhang et al. [146] studied the effect of long-term high-altitude hypoxia (a severe lack of oxygen) during gestation in sheep. Uterine arteries of pregnant sheep are affected by chronic hypoxia due to an inhibition effect of the large conductance Ca^2+^ activated K^+^ (BK_Ca_) channel activity by increasing oxidative stress. Treatment of the pregnant sheep with EUK-134 resulted in a mitigation of the hypoxia effects on BK_Ca_ channel currents in uterine arteries, alleviating pregnancy complications such as pre-eclampsia and fetal growth restriction.

The same manganosalen complex was used by Chen et al. [147] to protect ethanol-induced limb malformations in mice. In vivo treatment with EUK-134 resulted in diminished apical ectodermal ridge cell death as well as parallel reductions in the incidence and severity of limb defects in mouse fetuses (from 67.3% to 35.9%). The forelimb malformations were partially reversed by this manganosalen complex, including postaxial ectrodactyly, metacarpal, and ulnar deficiencies.

### 4.6. Adrenal and Liver Diseases

Since the imbalance between free radicals and antioxidants can be suffered by a variety of cells and issues, practically any organ can be affected, leading to a wide variety of pathologies. Kidney and liver function can be altered by excessive ROS, and again, manganosalen complexes appear as antioxidant therapeutic alternatives.

Kregel et al. [148] used EUK-189 to prevent age-related oxidative damage associated with environmental stress. They reported that this catalytic antioxidant blocked the activation of activator protein-1 (a redox-sensitive early response transcription factor involved in the regulation of cellular stress responses) and enhanced stress tolerance in aged animals by reducing cellular oxidative stress and subsequent accrual of hepatic injury in Fischer 344 rats. Yazdanparast et al. [149] reported the amelioration of diet-induced nonalcoholic steatohepatitis in rats by EUK-8 and EUK-134. These two compounds had hepatoprotective, hypolipidemic, hypocholestorolimic, and hypoglycemic effects on the in vivo model. Thus, the authors reported that EUK-8 and EUK-134 reduced the sera aminotransferases, the extent of lipid peroxidation, low density lipoprotein contents, cholesterol, and protein carbonylation. The same research group published another study about the protective effects of EUK-8, EUK-15, EUK-115, EUK-122, EUK-132, and EUK-134 against CCl_4_-induced damages in rats [150]. The manganosalen complexes ameliorated the effects of CCl_4_ by decreasing the levels of ROS, lipid and protein oxidations, and lipofuscin-like pigments formation on the liver and brain.

EUK-134 was used by Ghouleh et al. [151] to attenuate the vascular manifestations of sepsis in lipopolysaccharide-treated pigs. This catalytic antioxidant prevented the fall in renal blow flow, an effect associated with a decrease in nitrosative stress in the kidney supporting a renal protective effect.

### 4.7. Lifespan Extension

Harman proposed that organisms age because they accumulate oxidative damage that comes from ROS [152]. His free radical theory of aging prompted investigations to look for therapeutic antioxidants to lifespan extension. Although this theory was supported by later studies that demonstrate that increased production of ROS shortens lifespan [153] and that oxidative damage increases with age [154], other reports clearly contradict the basis of this theory [155]. The aim of this review is not to assess this controversy but to present the results of the different tests in this regard.

As already commented in the introductory section, Melov et al. [1] employed EUK-8 and EUK-134 to increase the mean lifespan of *Caenorhabditis elegans*. On the contrary, Hekimi et al. [156] reported that increased oxidative stress caused by deletion of the mitochondrial superoxide dismutase SOD-2 extended lifespan in the same nematode model. According to this latest study, decreased antioxidant function may extend lifespan. In vivo studies are sensitive to small changes in the environment, as Gems et al. [5] hypothesized to explain their results when they tried to reproduce the lifespan extension of the same nematode using EUK-8. Since they did not find any increase in lifespan upon treatment with this SOD mimetic, they conclude that this effect reported by Melov et al. should be very sensitive to subtle differences in the manner in which the manganosalen complex is administered. Their results raised doubts about the potential utility of EUK-8 in the therapeutic attenuation of aging. A subsequent study by Gems et al. [157] tested EUK-8 and EUK-134 again in *Caenorhabditis elegans*. In this study, they found that the synthetic mimetics elevated in vivo SOD activity levels (increases 5-fold) and exhibited protection when the worms were treated with superoxide generators (paraquat and plumbagin). Thus, the manganosalen complexes increased lifespan in nematodes compared to the control study, where superoxide levels were elevated, but they did not retard aging in the absence of superoxide generators.

EUK-8, EUK-134, and *Caenorhabditis elegans* met again in the study of Lithgow et al. [158]. They reported that these manganosalen complexes extended the lifespan of the worms and conferred resistance to two types of oxidative stress-inducing agents (paraquat and thermal stress). The protective effects of EUK-8 and EUK-134 were independent of insulin/IGF-I signaling, and they did not show any detrimental repercussion on development or fertility. Definitely, in vivo evaluation of these synthetic mimetics in aging studies involves careful consideration of complex concentration, complex delivery, and biotic environment.

Partridge et al. [159] treated *Drosophila melanogaster*, the fruit fly, with EUK-8 and EUK-134, reporting the antioxidant protective effects to rescue pathologies associated with elevated oxidative stress in SOD-deficient flies or normal flies exposed to induced oxidative stress. However, the synthetic antioxidants did not extend lifespan in normal, wild-type animals. In a different study, Sohal et al. [160] did not find lifespan extension administrating the EUK-8 to *Musca domestica*, both under normoxic and hyperoxic conditions.

In general, the life expectancy of animal models subjected to oxidative stress or with pathologies associated with this stress is increased with treatment with manganosalen complexes, as reviewed in previous sections (Table 5). However, the use of these synthetic antioxidants in normal and healthy individuals probably does not have any advantageous effect on lifespan extension.

## 5. Conclusions

Throughout the previous sections, the protective effects of manganosalen complexes to combat oxidative stress and its associated pathologies have been presented and discussed. These compounds showed efficiency to reverse different oxidative damage: neurodegenerative, inflammatory, cardiovascular, adrenal and liver diseases, skin damage, and fetal malformations (Figure 8). The increase of lifespan has been also reported for organisms exposed to oxidative stress, although far from being any elixir of life. Not all compounds have the same activity, as they show different lipophilicities, redox properties, or steric hindrance.

Of the nearly sixty in vivo trials, only a few of them did not give beneficial effects for the desired quality. Thus, Wada et al. [161] reported that EUK-134 was ineffective in promoting the restoration of prolonged low-frequency force depression, a state which may suffer skeletal muscles under vigorous activity. In this case, the antioxidant catalyst showed a positive effect on sarcoplasmic reticulum Ca^2+^ release in Wistar rats but a negative effect on myofibrillar Ca^2+^ sensitivity. On the other hand, Espósito et al. [162] reported the toxicity of EUK-108 (20–100 µM dose) toward *Danio rerio* individuals (zebrafish), particularly in terms of brain damage. Finally, Vanfleteren et al. [11] found that administration of EUK-8 to starving *Escherichia coli* cells surprisingly enhanced the production of ROS, resulting in a massive increase of oxidative damage.

Much is still unknown about the way these manganosalen complexes work in different pathologies. Drugs often have unrecognized effects, so we cannot be confident that the beneficial effects of these compounds were due solely to their antioxidant activities. More research is needed, particularly because of the properties showed by this type of compound that also varies with minor structural modifications. Evaluation in animal models continues to be necessary, since it cannot be assumed that the in vivo antioxidant activity of different analogues is similar. In the search to identify the best candidate, an orally available one would be the most suitable. Efforts should be directed towards obtaining an effective antioxidant, modulating its lipophilicity and redox potentials, that can be administered orally.

Despite the protective effect against ROS shown by these compounds in oxidative stress models in vitro and in vivo, their practical application in humans remains highly challenge. Clinical trials are crucial to assess the efficacy of this approach, especially after the results given by other antioxidants. Thus, although coenzyme Q10, β-carotene, α-tocopherol, or other antioxidant supplements showed highly encouraging results in in vivo animal models, most of their clinical trials in humans failed to reproduce positive results [163,164,165]. The synthetic catalytic antioxidants approach may have a plus compared with nonenzymatic antioxidants: They may regulate ROS by mimicking the mechanism of the native enzymes.

Finally, in vivo evaluation of pharmacological responses may be affected by multiple factors. Moreover, sometimes, the results are simplified by describing them as beneficial or harmful when, as declared by Paracelsus, the dose makes the medicine. ROSs play physiological roles in cell signaling and in the control of gene expression, processes that could be affected by antioxidant therapies. The catalytic oxidants should be administered just in the right dose to combat excessive ROS levels. In vitro studies with different cell cultures have shown higher activities for manganosalen complexes at lower doses than those used in the in vivo tests [6,23]. The antioxidant activity observed in cells can even decrease with increasing concentration [23], leading to curves that are not dose dependent. This behavior has been also reported in natural compounds such as curcumin or resveratrol, which present antioxidant effects at low doses but induce oxidative stress and cell death at high concentrations [166,167]. In this way, in vitro studies indicate that manganosalen complexes could also interact with other cellular pathways at high concentrations or with a receptor that could be suffering a threshold effect, that is, it would present higher affinity at low concentrations and would be desensitized at higher doses [168]. In this regard, one of the challenges for the translation of these antioxidant synthetic catalysts to animal or human studies is the use of drug carriers to effectively reach the target site at the appropriate doses [169]. Interaction with nanocarriers or conjugation of simple manganese complexes to synthetic polymers or proteins represent current and future avenues of research to translate manganosalen complexes to clinical applications.

## Figures and Tables

**Figure 1 antioxidants-09-00727-f001:**
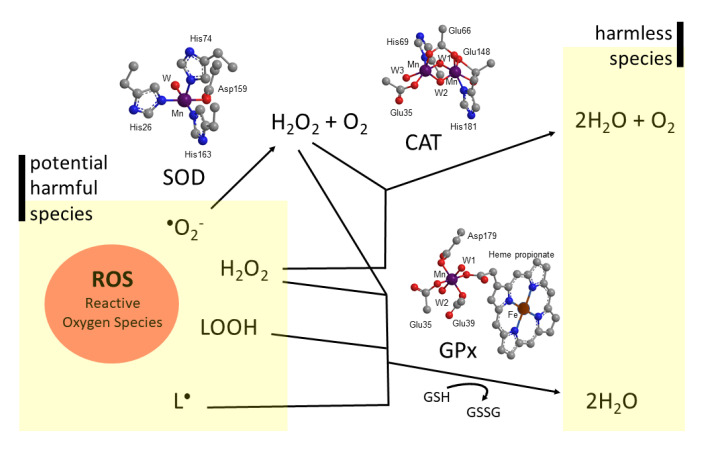
Chemical transformations of some of the reactive oxygen species (ROS) from potentially harmful species (superoxide radical, O_2_^•−^; hydrogen peroxide, H_2_O_2_; lipid hydroperoxides, LOOH; or lipid radicals, L^•^) into harmless species through manganese antioxidant enzymes: the figure shows the core of the active site of human mitochondrial SOD2, the core of the active site in *Lactobacillus plantarum* catalase (CAT), and the core of the active site of manganese glutathione peroxidase (GPx). W = water; GSH = monomeric glutathione; GSSG = glutathione disulfide.

**Figure 2 antioxidants-09-00727-f002:**
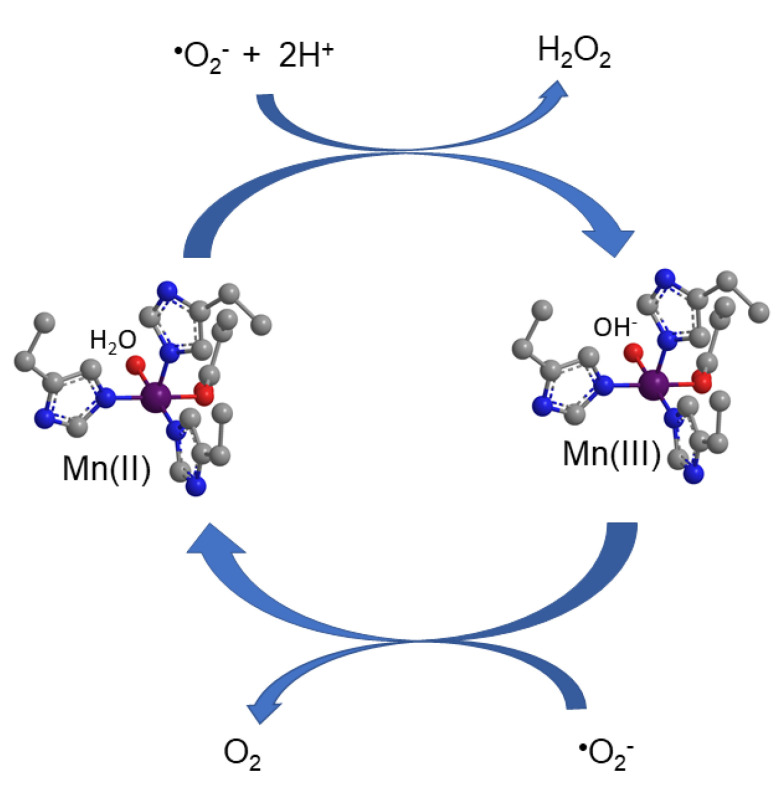
Mechanism of superoxide dismutase activity for superoxide radical anion disproportionation by the human manganese SOD2 enzyme.

**Figure 3 antioxidants-09-00727-f003:**
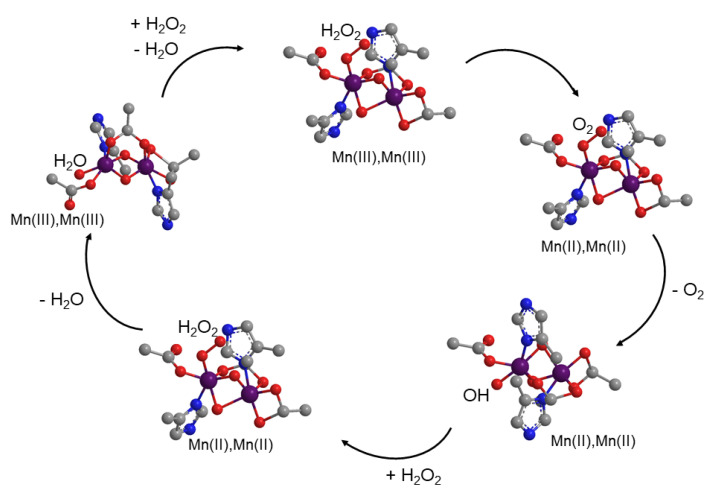
Mechanism of catalase activity for hydrogen peroxide disproportionation by manganese peroxidase enzyme from *Lactobacillus plantarum*.

**Figure 4 antioxidants-09-00727-f004:**
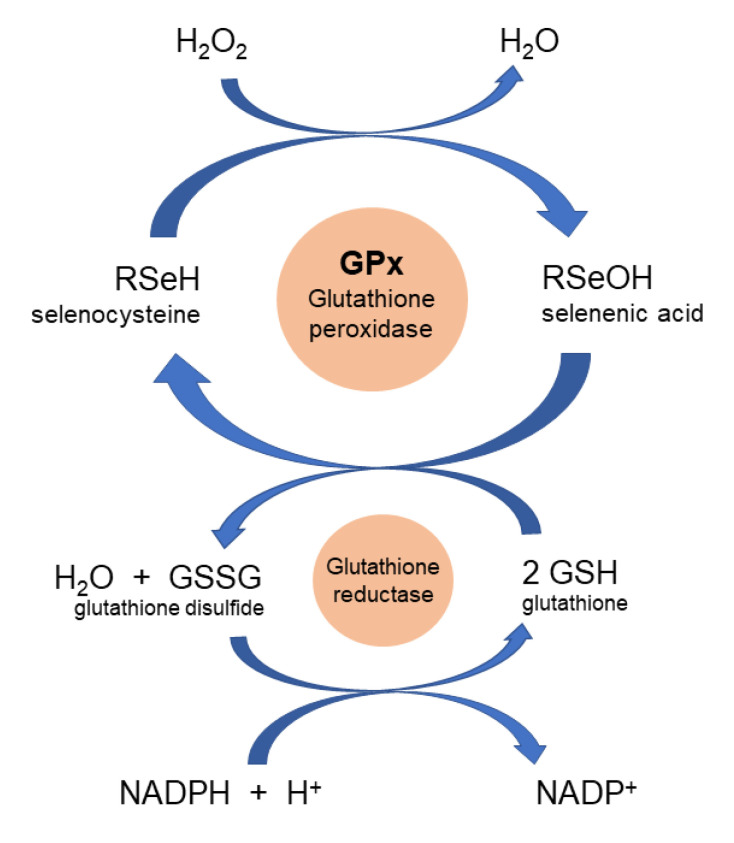
Mechanism of peroxidase activity for hydrogen peroxide reduction by the native glutathione peroxidase enzyme.

**Figure 5 antioxidants-09-00727-f005:**
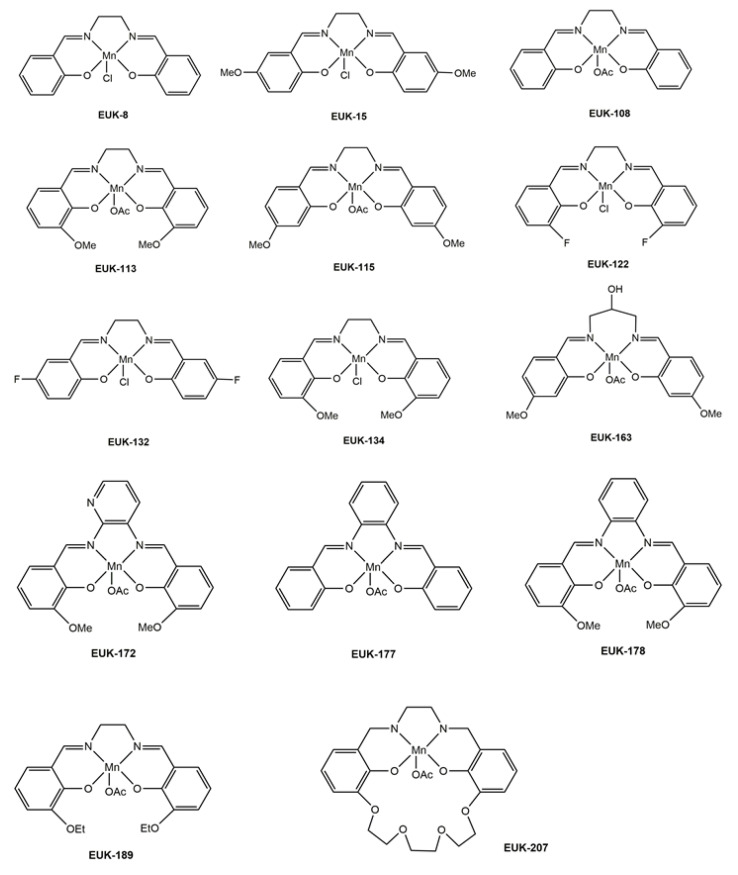
Structures of some manganosalen complexes, corresponding to the EUK series, with pharmacological relevance.

**Figure 6 antioxidants-09-00727-f006:**
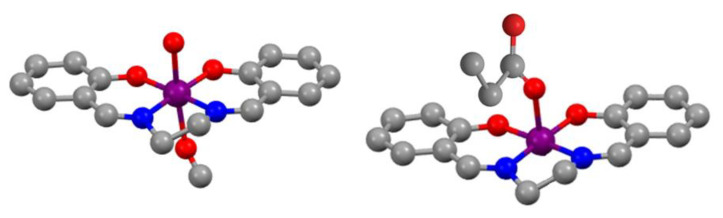
Manganosalen model complexes with distorted octahedral geometry (left, coordination number 6) or distorted square-planar pyramidal geometry (right, coordination number 5).

**Figure 7 antioxidants-09-00727-f007:**
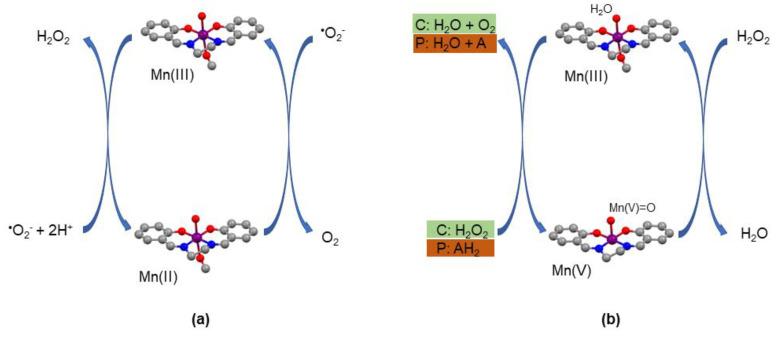
(**a**) Mechanism of superoxide dismutase activity for superoxide radical anion disproportionation by manganosalen complexes and (**b**) proposed mechanism of catalase (C: highlighted in green) or peroxidase (P: highlighted in brown) activities by manganosalen complexes (AH_2_ is an oxidizable substrate in peroxidase function).

**Figure 8 antioxidants-09-00727-f008:**
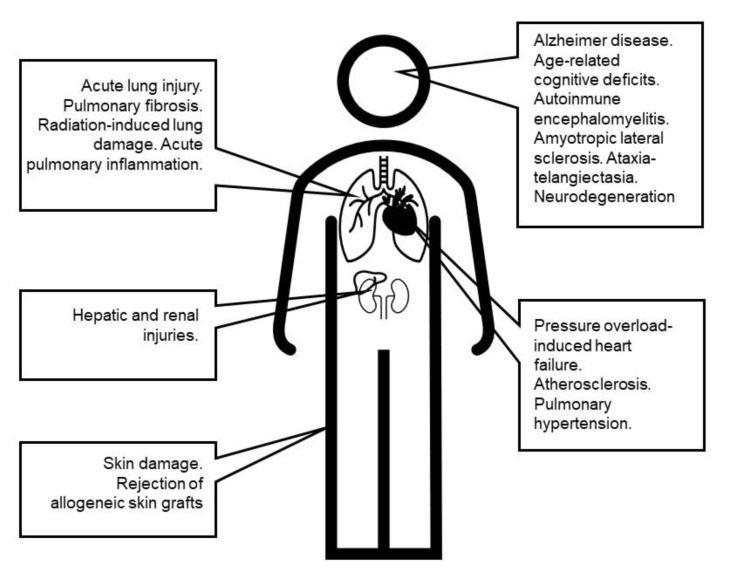
Some of the oxidative stress-related diseases tested in in vivo animal models with manganosalen complexes.

**Table 1 antioxidants-09-00727-t001:** Selected in vivo trials of manganosalen complexes treatments for combating neurodegenerative diseases.

Antioxidant Model	Disease	Animal Model	Dose	Outcomes	Ref.
EUK-8	Spongiform neurodegenerative disorder	SOD2 nullizygous mice	30 mg/kg	Rescue of the neurodegenerative disorder	[4]
EUK-134					
EUK-189					
EUK-207	Age-related cochlear cell degeneration	SAMP8/SAMR1 mice	0.2 mg/kg/day	Prevention of age-related hearing loss	[22]
EUK-8	Ischemia/reperfusion injury	Sprague-Dawley rats	1 mg/kg, 2 times	Protective effects	[91]
EUK-134	Kainite-induced neuropathology	Sprague-Dawley rats	10 mg/kg, 2 times	Prevention of excitotoxic neuronal injury	[101]
EUK-8	Autoimmune encephalomyelitis	Guinea pig MBP	100 mg/kg	Prevention and suppression of the disease	[102]
EUK-8	Neurodegeneration	SOD2 nullizygous mice	30 mg/kg	Rescue of neuronal cell death	[103]
EUK-189	Alzheimer disease	Tg2576 mice	30 mg/kg, 3 t/w	Amelioration of cataracts in the lenses	[104]
EUK-189	Loss of learning and memory function	C57BL/6N Sim mice	9 nmol/day-0.09 µM/day	Reversion of cognitive deficits	[106]
EUK-207					
EUK-189	Age-related cognitive deficits	C57BL/6N Sim mice	15–16 µg/Kg/day	Reduc. ^3^ of the age-related cognitive impairment	[107]
EUK-207					
EUK-8	Amyotrophic lateral sclerosis	SOD1-G93A mice	33 mg/kg, 3 t/w	Prolongation of survival	[109]
EUK-134					
EUK-8	Ischemic brain injury	Sprague-Dawley rats	30 mg/kg, 3 t/w	Reduction of brain infarct size	[110]
EUK-134					
EUK-134	Dopaminergic neurons death by neurotoxic	C57BL/6 mice	15 mM 1 day prior to the toxic	Attenuated the loss of nigral dopamine neurons	[111]
EUK-139					
EUK-189	Human prion disease	Balb/c mice	30 mg/kg, 3 t/w ^1^	Modest prolong. ^2^ survival. Reduc. ^3^ in oxidative damage to proteins.	[112]
EUK-189	Ataxia-telangiectasia	Mice lacking ATM gene	1.2 mg/kg/day	Correction of the neurobehaviroral abnormality	[113]
EUK-207	Radiation-induce cognitive impairments	C57Bl6/J mice	0.2 mg/kg/day	Mitigation of the cognitive injury	[114]

^1^ t/w: times a week; ^2^ prolong.: prolongation; ^3^ reduc.: reduction.

**Table 2 antioxidants-09-00727-t002:** Selected in vivo trials of manganosalen complexes treatments for combating inflammatory diseases.

Antioxidant Model	Disease	Animal Model	Dose	Outcomes	Ref.
EUK-189	Radiation-induced lung injury	Sprague–Dawley rats	30 mg/kg	Reduction micronucleous formation in lung fibroblasts	[119]
EUK-134	Proinflammatory damage in the diaphragm muscle	Mdx mice	30 mg/kg/day	Reduction of muscle damage	[120]
EUK-134	Acute pulmonary inflammation	C57BL/6 mice	200 µL aliquots of 3.2 mg total lipid at 2000 CPM/µL	Alleviate acute pulmonary inflammation	[121]
EUK-134	Paraquat-induced inflammation	Wistar rats	10 mg/kg	Protection from the toxic effects	[122]
EUK-207	Radiation-induced lung injury	Fisher rats	8 mg/kg	Mitigation of lung injury	[123]
EUK-207	1918 influenza virus	BALB/c mice	30 μg/day	Reduction of severity lung injury	[124]
EUK-8	Acute lung injury	Pigs	10 mg/kg bolus and 3 mg/kg.h, *n* = 6	Alleviate acute lung injury	[96,126,127]
EUK-207	Radiation-induced lung damage	Sprague–Dawley rats	8 mg/kg/day	Limitation of pulmonary fibrosis	[129,130]
EUK-207	Radiation-induced lung injury	Sprague–Dawley rats	8 mg/kg/day	Mitigation of the radiation effects	[131]

**Table 3 antioxidants-09-00727-t003:** Selected in vivo trials of manganosalen complexes treatments for combating cardiovascular diseases.

Antioxidant Model	Disease	Animal Model	Dose	Outcomes	Ref.
EUK-8	Pressure overload-induced heart failure	B6CBA mice hemizygous or homozygous for the X-linked Hq mutation	25 mg/kg/day	Prevention myocardial damage. Attenuation cardiac hypertrophy and fibrosis	[134]
EUK-189	Gamma irradiation	C3H/HeN and CD2F1 mice	70 mg/kg	Increase of 30-day survival	[135]
EUK-134	Pulmonary hypertension	Wistar rats	3 mg/kg/day	Prevention of diaphragm muscle weakness	[137]
EUK-207	Atherosclerosis	C57B1/6 mice	1 mg/kg/day	Reduction for endothelial-associated events	[138]

**Table 4 antioxidants-09-00727-t004:** Selected in vivo trials of manganosalen complexes treatments in ROS-mediated skin damage.

Antioxidant Model	Disease	Animal Model	Dose	Outcomes	Ref.
EUK-8	Rejection of allogeneic skin grafts	BALB/c and C57BL/6 mice	25 mg/kg/day	Attenuation on graft rejection	[140]
EUK-134					
EUK-189					
EUK-134	UV-induced skin damage	Humans volunteers	Topical 0.01–0.1%, 3 µL cm^−2^	Protection of skin surface from accumulating oxidative damage	[12]
EUK-207	Radiation dermatitis	WAG/RijCmcr mice	1.8 mg/kg/day	Promotion wound healing in irradiated skin	[143]
EUK-207	Radiation-induced DNA damage or lipid peroxidation	C3H/HeJ mice	30 mg/kg	Protection before irradiation. Mitigation of lipid peroxidation	[144]

**Table 5 antioxidants-09-00727-t005:** Selected in vivo trials of manganosalen complexes treatments for lifespan extension.

Antioxidant Model	Animal Model	Dose	Outcomes	Ref.
EUK-8	*Caenorhabditis elegans* nematode	0.05 mM	Increase in mean-life span of 44%	[1]
EUK-134				
EUK-8	*Caenorhabditis elegans* nematode	0.05–5 mM	No increase in lifespan	[5]
EUK-8	*Caenorhabditis elegans* nematode	0.25–0.5 mM	Increase lifespan in presence of superoxide generators	[157]
EUK-134				
EUK-8	*Caenorhabditis elegans* nematode	0.05–1 mM	Increase lifespan in presence of superoxide generators	[158]
EUK-134				
EUK-8	*Drosophila melanogaster* Fruit fly	0.025–0.5 mM	No extension lifespan in normal animals	[159]
EUK-134				
EUK-8	*Musca domestica* fly	0.025–0.5 mM.	No extension lifespan	[160]

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
