# Peer review of "Pursuing the Elixir of Life: In Vivo Antioxidative Effects of Manganosalen Complexes"

_antioxidants, 2020, doi:10.3390/antiox9080727_

Round 1

Reviewer 1 Report

This review provides a well presented and comprehensive coverage of manganosalen complexes from function to in vivo roles. The narrative was clear to follow yet quite detailed.  The document was a pleasure to read and is a valuable contribution to the field. However there are a number of minor edits that could help improve the work further.

Title - change to: "Pursuing the Elixir of Life: In vivo antioxidative effects of manganosalen complexes"

line 90. Clarify the aims and primary focus of the review. The aim stated here  differs from the aim stated in the abstract - reconcile for consistency.

There does not appear to be sufficient discussion regarding dose-dependency of action, side effects, pro-oxidant phase of redox cycles, dose and interaction based toxicities of the manganosalen complexes. This is only slightly touched on late in the discussion/conclusions. The conclusion can be strengthened by focusing more on:  What are the current major limitations for translation to large animal and human studies? Is target specificity the problem? Which compounds hold more promise/which target/disease? These points help set direction for future work. 

Author Response

REVIEWER: This review provides a well presented and comprehensive coverage of manganosalen complexes from function to in vivo roles. The narrative was clear to follow yet quite detailed.  The document was a pleasure to read and is a valuable contribution to the field. However there are a number of minor edits that could help improve the work further.

ANSWER: We would like to thank this reviewer for such encouraging feedback. We are pleased with the positive evaluation of the manuscript, giving us very valuable suggestions and comments. Here we address each question raised by this reviewer, one by one.

REVIEWER: Title - change to: "Pursuing the Elixir of Life: In vivo antioxidative effects of manganosalen complexes"

ANSWER: Thank you for the suggestion. It has been done, as suggested by the reviewer.

REVIEWER: line 90. Clarify the aims and primary focus of the review. The aim stated here differs from the aim stated in the abstract - reconcile for consistency.

ANSWER: The sentence of line 90 has been modified according to the referee’s comment to avoid the confusion about the primary focus of the review.

REVIEWER: There does not appear to be sufficient discussion regarding dose-dependency of action, side effects, pro-oxidant phase of redox cycles, dose and interaction based toxicities of the manganosalen complexes. This is only slightly touched on late in the discussion/conclusions. The conclusion can be strengthened by focusing more on:  What are the current major limitations for translation to large animal and human studies? Is target specificity the problem? Which compounds hold more promise/which target/disease? These points help set direction for future work. 

ANSWER: Thank you for the comment. We agree with the comments proposed by the reviewer, these are key issues that will determine the future development of research in this field. We have added in section 5 (conclusions) the following paragraph regarding the reviewer’s suggestions:

“In vitro studies with different cell cultures have shown higher activities for manganosalen complexes at lower doses than those used in the in vivo tests [6,23]. The antioxidant activity observed in cells can even decrease with increasing concentration [23], leading to curves that are not dose dependent. This behaviour has been also reported in natural compounds such as curcumin or resveratrol, which present antioxidant effects at low doses but induce oxidative stress and cell death at high concentrations [169-170]. In this way, in vitro studies indicate that manganosalen complexes could be also interacting with other cellular pathways at high concentrations, or with a receptor that could be suffering a threshold effect, that is, it would present higher affinity at low concentrations and would be desensitized at higher doses [171]. In this regard, one of the challenges for the translation of these antioxidant synthetic catalysts to animal or human studies is the use of drug carriers to effectively reach the target site at the appropriate doses [172]. Interaction with nanocarriers or conjugation of simple manganese complexes to synthetic polymers or proteins represent current and future avenues of research to translate manganosalen complexes to clinical applications.”

Reviewer 2 Report

This review covers a highly intriguing topic in great depth, and with the following changes, the review could be improved:

  • In the first paragraph, some of the findings are overstated. For instance, a threefold increase in lifespan is unremarkable in an SOD2-null mouse. Those mice are lacking critical antioxidant functions, and providing antioxidant treatment merely reduces the severity of their oxidative stress. There is no direct relevance to the natural lifespan.
  • Superoxide radical is referred to as “harmful.” Yes, but it is also regularly cited as signaling molecule, so ultimately its biological role is more complex.
  • The manuscript would benefit from greater focus. There is a considerable amount of content related to general antioxidant pathways, which may not be necessary.
  • Figures 7, 8, and 9 could be condensed/merged to save space.
  • A very thorough effort, but it would be improved by narrowing the focus and trimming content. Currently, the manuscript is sprawling, and at times, does not clearly articulate its own purpose.

Author Response

REVIEWER: This review covers a highly intriguing topic in great depth, and with the following changes, the review could be improved:

ANSWER: We would like to thank this reviewer for the careful reading of the manuscript and for his/her opinions and suggestions, which have certainly contributed to improve the quality of this article. We have modified the manuscript following the suggestions of this referee as outlined below.

REVIEWER: In the first paragraph, some of the findings are overstated. For instance, a threefold increase in lifespan is unremarkable in an SOD2-null mouse. Those mice are lacking critical antioxidant functions, and providing antioxidant treatment merely reduces the severity of their oxidative stress. There is no direct relevance to the natural lifespan.

ANSWER: The discussion on this study (ref. 4) has been modified to incorporate the reviewer’s point: “When the mice were injected with the manganosalen complexes…they lived three times longer, indicating that these artificial complexes may act as synthetic mimetics of SOD in an animal model lacking critical antioxidant functions, and therefore, without direct relevance to the natural lifespan”. Additionally, the reference of this study has been removed from table 5 (selected in vivo trials of manganosalen complexes treatments for lifespan extension), and section 4.7 (lifespan extension) has also been modified according to the reviewer’s comment.

REVIEWER: Superoxide radical is referred to as “harmful.” Yes, but it is also regularly cited as signaling molecule, so ultimately its biological role is more complex.

ANSWER: We agree with the reviewer’s clarification. The idea of the beneficial role of ROS was previously stated along the manuscript. See, for instance:

Section 2: “ROS may play a role in signaling functions, consequently activating protective and adaptative programs [29-30]”

Section 5 (Conclusions): “ROS play physiological roles in cell signaling and in the control of gene expression, processes that could be affected by antioxidant therapies”

However, excessive ROS levels cause harmful effects. To reconcile this divergence, we have added the adjective "potential" to that of "harmful" in different sections of the manuscript, including figure 1.

REVIEWER: The manuscript would benefit from greater focus. There is a considerable amount of content related to general antioxidant pathways, which may not be necessary.

ANSWER: Thank you for the suggestion. We have shortened section 2 (manganese superoxide dismutases, peroxidases and catalases), removing part of the discussion on the structure of native antioxidant enzymes and their reaction mechanisms.

REVIEWER: Figures 7, 8, and 9 could be condensed/merged to save space.

ANSWER: Done as suggested by the reviewer.

REVIEWER: A very thorough effort, but it would be improved by narrowing the focus and trimming content. Currently, the manuscript is sprawling, and at times, does not clearly articulate its own purpose.

ANSWER: We have tried to narrow the focus of the manuscript by removing structural details about antioxidant enzymes and their general pathways of action (section 2). The discussion about the structural features that enhances the antioxidant activity of manganosalen complexes has also been shortened (section 3). As commented before, the figures 7, 8 and 9 have been merged into a single figure. We think that these modifications have narrowed the focus of the manuscript, as suggested by the referee, keeping the narrative thread of the review.

Round 2

Reviewer 2 Report

All issues have been addressed.